# Integrative Transcriptomic and Metabolomic Analysis of Muscle and Liver Reveals Key Molecular Pathways Influencing Growth Traits in Zhedong White Geese

**DOI:** 10.3390/ani15091341

**Published:** 2025-05-06

**Authors:** Kai Shi, Xiao Zhou, Jiuli Dai, Yuefeng Gao, Linna Gao, Yangyang Shen, Shufang Chen

**Affiliations:** 1Ningbo Academy of Agricultural Sciences, Institute of Livestock and Poultry Research, Ningbo 315040, China; shikai2024@126.com (K.S.);; 2College of Applied Engineering, Henan University of Science and Technology, Sanmenxia 472000, China; nexhyf@hotmail.com; 3Institute of Animal Science, Jiangsu Academy of Agricultural Sciences, Nanjing 210014, China; 4School of Life Sciences and Food Engineering, Hebei University of Engineering, Handan 056038, China; 5Key Laboratory of Crop and Animal Integrated Farming, Ministry of Agriculture and Rural Affairs, Nanjing 210014, China

**Keywords:** geese growth, breast muscle, liver, transcriptome, metabolome

## Abstract

This study explored the growth mechanisms of geese by analyzing transcriptomic and metabolomic differences between high-weight (HW) and low-weight (LW) Zhedong White geese. The HW geese exhibited significantly higher body weight and average daily gain than LW geese, with no significant difference in the yields of certain organs. In the breast muscle, 19 differentially expressed genes (DEGs) related to pathways like PPAR signaling and fatty acid biosynthesis were found, along with 59 differential accumulation metabolites (DAMs) affecting glutathione and vitamin B6 metabolism. In the liver, 106 DEGs and 202 DAMs were identified, influencing various metabolic pathways. Correlation analysis revealed links between specific gene expressions and metabolite levels. These findings enhance the understanding of goose growth and provide molecular markers for breeding.

## 1. Introduction

Geese (*Anser cygnoides*) are important economic domestic poultry with a long history of domestication and abundant diversity, raised in most parts of the world [1,2]. Goose meat is an excellent animal protein source, containing all kinds of amino acids that can meet daily basic nutritional demands [3]. Goose eggs are not only a rich source of protein that is easily absorbed, but they also contain a significant amount of unsaturated fatty acids, which can aid in lowering cholesterol levels and preventing cardiovascular diseases [4]. In addition, foie gras and goose down are also of high economic value; foie gras is widely favored by diners and goose down is an excellent insulation material. Geese are raised on a huge scale in China and, as the third-largest poultry category in 2020, the total production of goose meat is up to 4.28 million tons in 2021 [5]. Until now, the breeding progress of geese has lagged behind that of chickens and ducks, primarily due to the lack of advanced breeding technologies, which hinders the overall development of the goose industry. Therefore, exploring the molecular mechanism and excavating key genes that influence goose growth is an effective method to resolve the current predicament.

Skeletal muscle is an essential organ for motion and energy metabolism, accounting for about 40% of the animal’s body weight [6], and previous studies found that the weight of breast and thigh muscles account for 34% of body weight among different geese breeds [7]. Improving meat quantity and quality is an important direction in livestock. Skeletal muscle development is an elaborate process that is regulated by multi-spatial-temporal gene expression, including *MyoD*, *MyoG*, *Myf5*, and other regulatory factors [8]. Through whole-transcriptome sequencing, *LAMA4*, *SLC39A13*, *TRAF2*, *PLEKHM3*, and *PPP2R5A* were key genes to affect geese myoblast differentiation, and several circRNA-miRNA-mRNA networks synergistically regulate myogenesis in geese [9]. The liver is the largest visceral organ, which decomposes red blood cells, deposits lipids, regulates glycogen storage, and produces hormones [10]. Abundant proteins were synthesized in the liver, which are responsible for detoxification and immunity [11]. It has been reported that increased levels of antioxidative genes in goose liver is the key factor to affect body weight [12]; other studies found that hepatic glucose metabolism was necessary for geese growth [13]. Furthermore, there is a complex interaction between muscle and liver; muscle loss was found in chronic liver patients due to disturbed GH/IGF-1 axis and nitrogen homeostasis in the liver [14]. Hence, the muscular and hepatic state directly influences the animal’s growth.

The Zhedong White goose is a renowned native breed in China, characterized by its rapid growth rate and raised in 19 provinces. It serves as an excellent model for exploring the mechanisms underlying goose development. In this study, we utilized transcriptomic and metabolomic analyses to identify key muscular and hepatic genes and metabolites associated with growth differences in geese. This research provides new insights into the growth of geese and identifies potential targets for molecular-assisted selection in future breeding efforts.

## 2. Materials and Methods

### 2.1. Samples

Five-hundred male Zhedong White geese were reared in a single batch under uniform conditions. Geese were housed in groups of six per cage within a closed barn, with goslings having ad libitum access to food and water. During the rearing period, the humidity was kept between 65% and 70%. For temperature control, goslings aged 1–3 weeks were initially kept at 33 °C, with the temperature decreased by 1 °C daily until reaching the outdoor temperature. Goslings aged 4–10 weeks were reared at the outdoor temperature. In terms of photoperiod management, different age groups of goslings were exposed to different durations and intensities of light: goslings aged 1–3 weeks received 24 h of light per day at an intensity of 20–30 lux/m^2^, those aged 4–6 weeks were given 16–18 h of light daily at 5–10 lux/m^2^, and goslings aged 7–10 weeks had 12–14 h of light daily at 5–10 lux/m^2^. Their body weight (BW) was measured after a 12 h fasting period at both 1 and 70 days of age. The nutritional composition of geese is detailed in Table 1 [15]. Based on the body weight at 70 days, four healthy geese with high body weight and four with low body weight were selected for slaughter via jugular vein bleeding. The average daily gain (ADG) was calculated by subtracting the body weight at 1 day of age from the body weight at 70 days of age and dividing the result by the total number of days between these two time points. The percentage of half-eviscerated weight was determined by taking the ratio of the post-slaughter weight, after removal of the head, feet, wing tips, blood, gastrointestinal contents, and part of the viscera, to the pre-slaughter body weight. Similarly, the percentage of eviscerated weight was calculated based on the weight following the removal of the head, feet, wing tips, blood, gastrointestinal contents, all viscera, and all glands, again expressed as a percentage of the pre-slaughter body weight. For the yields of specific tissues including the head, brain, breast muscle, heart, liver, gallbladder, glandular stomach, and pancreas, they were calculated as the ratio of each respective tissue weight to the pre-slaughter body weight and expressed as a percentage. The breast muscle and liver were then isolated and quickly frozen using liquid nitrogen after weighing.

### 2.2. Transcriptome Sequencing

Total RNA of muscle and liver was extracted through Trizol regent (Invitrogen, Carlsbad, CA, USA) following the manufacturer’s instruction; the RNA purity and quality were measured using Nanodrop 2000 spectrophotometer (Thermo Scientific, Waltham, MA, USA) and Agilent 2100 Bioanalyzer (Agilent Technologies, Palo Alto, CA, USA). mRNA purification and library construction were based on Ribo-ZeroTM Magnetic Kit (Epicentre, Madison, WI, USA) and Hieff NGS^®^ Ultima Dual-mode RNA Library Prep Kit (Yeasen, Shanghai, China). The paired-end sequencing method was adopted and used the Illumina Nova-Seq 6000 sequencing platform by Novogene Biotechnology Co. (Beijing, China). Acquired raw sequencing data were filtered through Fastp software (v0.21.0), and clean data were aligned with the geese reference genome (GCA_000971095.1) using Hisat2 (v2.2.1) [16]. The DESeq2 package (v1.42.0) was used to detect differentially expressed genes (DEGs) (adjusted *p*-value < 0.05 and |log2 fold change| > 1.0) in muscle and liver between geese with low body weight and those with high body weight [17]. ClusterProfiler (v4.10.0) was employed to analyze the Kyoto Encyclopedia of Genes and Genomes (KEGG) pathway [18].

### 2.3. Metabolomic Analysis

Metabolites from the muscle and liver of Zhedong White geese were extracted by grinding the tissues with liquid nitrogen and resuspending them in prechilled 80% methanol. Equal volumes of metabolites from each sample were combined to create a quality control (QC) sample, while blank samples were prepared to remove background ions. Metabolite analysis was performed using an LC-MS/MS system in both positive and negative ion modes for relative quantification. Metabolites were identified and quantified by matching retention times and mass-to-charge ratios (m/z) against reference databases. Metabolomic data were obtained after eliminating background ions using blank samples and normalizing the data. The identified metabolites were annotated using the KEGG (https://www.genome.jp/kegg/pathway.html, accessed on 12 November 2024), HMDB (https://hmdb.ca/metabolites, accessed on 20 December 2024), and LIPIDMaps (http://www.lipidmaps.org/, accessed on 28 December 2024) databases. Metabolites with a Variable Importance in Projection (VIP) score greater than 1, a *p*-value less than 0.05, and a fold change of ≥2 or ≤0.5 were considered differential metabolites. The functions of these identified metabolites were further explored using the online tool MetaboAnalyst (v6.0) [19].

### 2.4. Quantitative Real-Time PCR

Reverse transcription of 1 μg total RNA from each sample into cDNA was performed using the PrimeScript RT Master Mix (Takara, San Jose, CA, USA) following the manufacturer’s protocol. The reaction conditions were as follows: 10 min at 25 °C, 15 min at 37 °C, and 5 s at 85 °C. The resulting cDNA was stored at −20 °C. For quantitative real-time PCR (qRT-PCR) analysis on a Step-One Real-Time PCR System (Applied Biosystems, Waltham, MA, USA), a 20 μL reaction mixture was prepared containing 2 μL cDNA, 0.5 μL forward primer (10 μM), 0.5 μL reverse primer (10 μM), 10 μL 2× SYBR Premix ExTaq II (Takara), and 7 μL RNase-free H_2_O. The amplification protocol consisted of an initial denaturation at 94 °C for 30 s, followed by 39 cycles of 15 s at 94 °C and 30 s at 58–62 °C. Transcript expression levels were normalized to the housekeeping gene Gapdh, and results were analyzed using the ΔΔCt method to determine fold-change relative to the control. Primer sequences are provided in Table 2.

### 2.5. Statistical Analysis

Data were analyzed using Prism (v.8.0; GraphPad Software, San Diego, CA, USA) for Student’s *t*-test. Data were displayed as the mean ± standard deviation (SD). *p*-value < 0.05 indicated that the results were statistically significant.

## 3. Results

### 3.1. Growth Performance and Carcass Yield

A total of 500 Zhedong White geese were raised, and the top 1% of highest (HW) and lowest (LW) body weight geese at 70 days of age were collected. At 1 day of age, no significant difference in body weight was observed between the LW and HW geese. However, by day 70, body weight in the HW group was significantly greater (*p*-value < 0.05), and the average daily gain was also improved compared to the LW group (Table 3). The percentages of half-eviscerated and eviscerated weights in the HW group, along with the yields of the head, brain, breast muscle, and heart, were lower than those in the LW group. The yields of the liver, gizzard, glandular stomach, and pancreas were not changed between the two groups (*p*-value > 0.05).

### 3.2. Transcriptomic and Metabolomic Profiles of Breast Muscle

In comparison to the LW group, the HW group exhibited six up-regulated differentially expressed genes (DEGs), including COL11A2, COL22A1, and MBOAT4, and 13 down-regulated DEGs, such as TF, ANGPTL4, and SYNC in breast muscle (Appendix A). These DEGs were associated with the PPAR signaling pathway, adipocytokine signaling pathway, fatty acid biosynthesis, and ferroptosis (Figure 1A,B). The relative expression of selected DEGs suggested that the results from our RNA seq are accurate (Figure 1C). Additionally, a total of 59 differential metabolites were identified in the breast muscle between HW and LW geese (Appendix A), related to glutathione, riboflavin, vitamin B6, and biotin metabolism (Figure 1D–F). Correlation analysis indicated that the levels of COL11A2 and COL22A1 were positively correlated with the content of S-(2-Hydroxy-3-buten-1-yl)glutathione (Figure 1G).

### 3.3. Transcriptomic and Metabolomic Profiles of the Liver

In the liver, 43 up-regulated (i.e., THSD4, SLC25A30, SOX6, FGFRL1, and CNST) and 63 down-regulated (i.e., CREB3L3, NR1H4, LOC106049048, PCK1, and UAP1) DEGs were detected between the LW and HW groups (Appendix A), which were related to the TGF-beta signaling pathway, pyruvate metabolism, glycolysis/gluconeogenesis, and the adipocytokine signaling pathway through KEGG analysis (Figure 2A,B). The protein–protein interaction network of DEGs from the liver suggested that PCK1, TFRC, and JUN may be the pivotal genes in liver to affect goose development (Figure 2C). The relative level of identified DEGs in the liver indicated that our RNA-seq is credible (Figure 2D). Through metabolome analysis, a total of 202 differential metabolites (e.g., 5,6,7,8-Tetrahydro-2,4-dimethylquinoline, 2-Hydroxyhexadecanoic acid, and coumarin) were screened between two groups (Appendix A). These metabolites were involved in pyrimidine metabolism, phenylalanine, tyrosine and tryptophan biosynthesis, nitrogen metabolism, and phenylalanine metabolism (Figure 2E,F). Conjoint analysis of the top 20 DEGs and top 30 differential metabolites showed a strongly positive correlation in THSD4 and 2-Hydroxyhexadecanoic acid with 5,6,7,8-Tetrahydro-2,4-dimethylquinoline (Figure 2G).

### 3.4. The Interplay Between the Breast Muscle and Liver

To investigate potential liver–muscle crosstalk, we identified shared DEGs and DAMs between the liver and breast muscle. One DEG, LOC106049048, was down-regulated in both the liver and muscle of the HW group. Additionally, four DAMs—mogrol, brassidic acid, flabelline, and L-Leucyl-L-alanine—were found in both tissues (Figure 3). Notably, the relative contents of mogrol, brassidic acid, and flabelline were consistently increased between the muscle and liver in the HW group (*p*-value < 0.05).

## 4. Discussion

As consumer living conditions improve, geese have gained popularity for their meat, eggs, foie gras, and down, with better growth performance yielding increased economic benefits. However, the mechanisms underlying goose growth remain poorly understood. In this study, 500 Zhedong White geese were raised, and eight geese (four high-weight and four low-weight) were selected based on their body weight at 70 days. The body weight of the high-weight group was, on average, 748.75 g heavier than that of the low-weight group, and the average daily gain of the high-weight geese exceeded that of the low-weight geese by 10.77 g (*p*-value < 0.05). The yields of digestive organs (liver, gizzard, glandular stomach, and pancreas) displayed no significant difference between the LW and HW groups; the percentage of half-eviscerated weight and eviscerated weight was also shown to be stable between two groups, which may have resulted from the limited samples employed in this study. A previous study indicated that geese with higher feeding intake rates exhibited increased body weight and liver weight [20]. Thus, the enhanced yield of digestive organs in the high-weight geese may contribute to the improved digestion and absorption of nutrients, ultimately accelerating their growth rate.

Skeletal muscle is a vital organ for motion and metabolism, comprising approximately 40% of body weight. While the yield of breast muscle showed no significant difference between the HW and LW groups, the weight of breast muscle in the HW geese was indeed higher than that in the LW geese (*p*-value < 0.05). To explore the changes in breast muscle, transcriptomic and metabolomic analyses were performed to detect key genes and metabolites. In breast muscle, 19 differentially expressed genes were detected between the two groups, including *COL22A1*, *COL11A2*, and *ANGPTL4*. *COL22A1* encodes the synthesis of collagen XXII; the knockout of *COL22A1* in zebrafish resulted in a muscular dystrophy-like phenotype and exogenous COLXXII could rescue the defect [21]. Recent single-cell RNA-sequencing of skeletal muscle indicated that *COL22A1* located at the end of myofiber may influence muscle development by regulating the function of the neuromuscular junction [22,23]. *COL11A2* (collagen type XI alpha 2 chain) is pivotal for vertebral development, and was determined as a candidate gene for the body length of pigs [24], large yellow croakers [25], mice [26], and goats [27]. *ANGPTL4* is an angiopoietin-like protein that regulates fat metabolism; previous studies found that the body weight and fecal fat content were higher in *ANGPTL4*-deficient mice than in wild-type mice [28] and increased levels of *ANGPTL4* induced by linoleic acid inhibit myoblast differentiation through suppressing Wnt/β-catenin signaling pathway [29]. Through metabolomic analysis, the contents of 59 metabolites were significantly different in breast muscle between the HW and LW geese, and were involved in glutathione metabolism, vitamin B6 metabolism, riboflavin metabolism, and biotin metabolism. Conjoint analysis of DEGs and DAMs showed a strong positive correlation between S-(2-Hydroxy-3-buten-1-yl)glutathione and both *COL11A2* and *COL22A1*; the results indicated that *COL11A2* with *COL22A1* may influence skeletal muscle development by regulating glutathione metabolism.

In this study, a total of 99 DEGs (i.e., *THSD4*, *CREB3L3*, and *NR1H4*) of the liver were identified between HW and LW groups, and they were involved in the signaling pathway of cytokine–cytokine receptor interaction, pyruvate metabolism, glycolysis, and several regulations of the development process. *THSD4* (Thrombospondin Type 1 Domain Containing 4) is a protein-coding gene with a hydrolase activity that was identified as a candidate gene for body fatness in mice [30]. Previous suggested patients with THSD4 mutation showed skeletal defects [31], and the gene has also been identified as a candidate gene that influenced broilers’ growth and meat quality [32]. *CREB3L3* (cAMP-responsive element-binding protein 3-like 3) is a membrane-bound transcription factor that could be activated during fasting to regulate triglyceride metabolism, it has been reported that the increase in body weight from the *CREB3L3*-overexpressed mouse was significantly inhibited during high-fat and high-sucrose feeding [33]. Another study proved that the level of plasma triglycerides in CREB3L3 knockout mice was increased, and the deficiency in CREB3L3 activated the hepatic-proliferative function [34]. *NR1H4* could function as a receptor for hepatic bile acid, and its mutation changed the level of fasting glucose and free fatty acid [35]; other studies proved that *NR1H4* was a key regulated gene related to liver development and an upregulated level of *NR1H4* would inhibit lipid synthesis [36]. Through metabolomic analysis, 202 differential metabolites were detected in the liver between the HW and LW geese, and these were involved in pyrimidine metabolism, nitrogen metabolism, phenylalanine metabolism, arginine biosynthesis, and phenylalanine, tyrosine and tryptophan biosynthesis. Various amino acid biosynthesis in the liver is essential for animal growth. Through conjoint analysis of DEGs and DAMs, the level of *THSD4* was positively correlated with the content of 2-Hydroxyhexadecanoic acid. Former studies compared the differential metabolites between sheep with high body weight and those with low body weight and found that the content of 2-Hydroxyhexadecanoic acid was increased in the muscle and rumen content of sheep with high body weight [37,38].

*LOC106049048* (*ACAD11L*) is an acyl-CoA dehydrogenase family member 11-like protein, the shared DEG in the liver and the breast muscle. *ACAD11* and *ACAD10* regulate mammalian 4-hydroxy acid lipid catabolism, and previous studies found that the knockout of *ACAD11* slightly increased the body weight of mice [39], *ACAD10*-deficient mice showed enhanced mass of liver, epididymal fat, and tibialis anterior muscle [40]. Therefore, the down-regulated expression of *LOC106049048* may promote the geese’s growth by regulating the lipid metabolism of the geese’s skeletal muscle and liver. Compared with the LW group, the contents of mogrol, brassidic acid, and flabelline were increased in liver and breast muscle in the HW group. Mogrol is a pharmacologically active ingredient isolated from the fruits of *Siraitia grosvenorii* that could relieve the development of pulmonary fibrosis and hyperglycemia [41,42]. The trans-isomer of erucic acid is brassidic acid, which is a bioactive substance that has antimicrobial action [43]. Based on these findings, we hypothesized that geese’s growth was potentially influenced by altering the metabolism of breast muscle and liver.

## 5. Conclusions

In this study, we systematically identified the differential genes and metabolites in the liver and breast muscles of geese with varying growth performances. Our findings enhance the understanding of the transcriptomic and metabolomic basis underlying high and low growth performance in Zhedong White geese. The potential genes and metabolites identified can serve as selection markers in breeding programs aimed at promoting growth performance, although further functional validation is necessary in subsequent studies.

## Figures and Tables

**Figure 1 animals-15-01341-f001:**
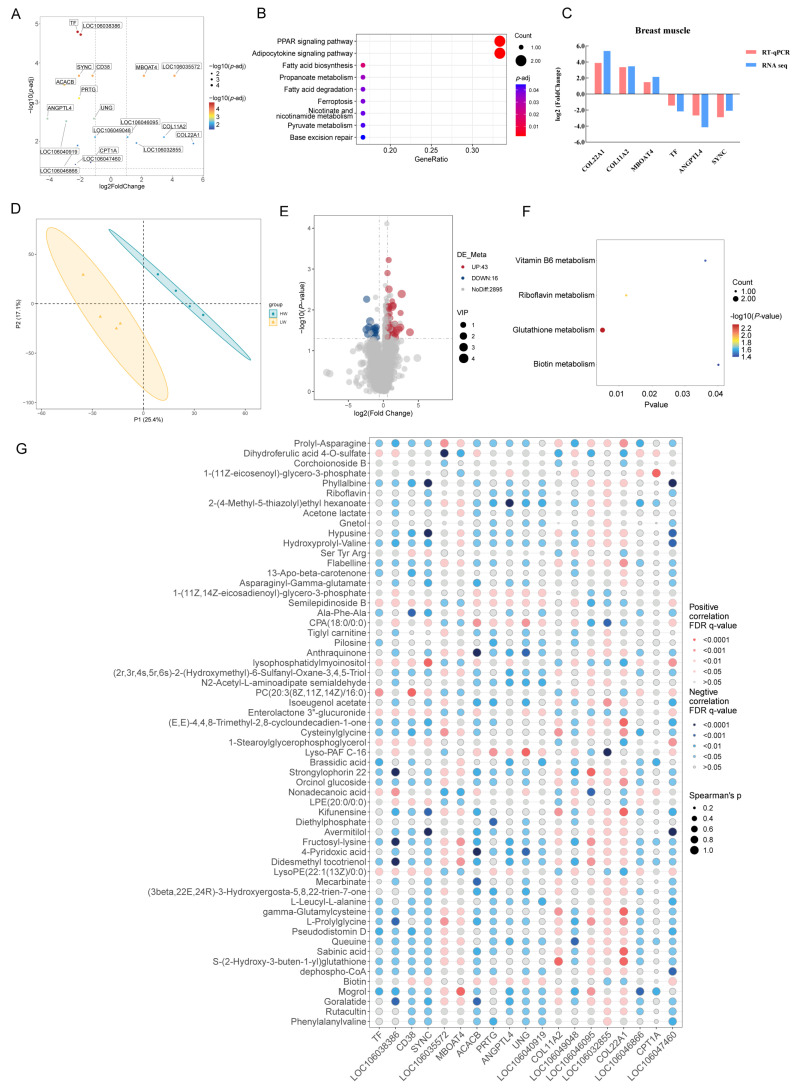
Transcriptomic and metabolomic analysis of the breast muscle of Zhedong White geese with high and low body weight. (**A**) Volcano plot of DEGs in the breast muscle between the HW and LW geese. (**B**) KEGG analysis of DEGs in the breast muscle. (**C**) Validation of DEGs from breast muscle by quantitative real-time PCR. (**D**) Principal component analysis of metabolomic analysis in breast muscle. (**E**) Volcano plot of DAMs in the breast muscle of the HW and LW groups. (**F**) Functional analysis of DAMs in the breast muscle. (**G**) The conjoint analysis of DEGs and DAMs in the breast muscle.

**Figure 2 animals-15-01341-f002:**
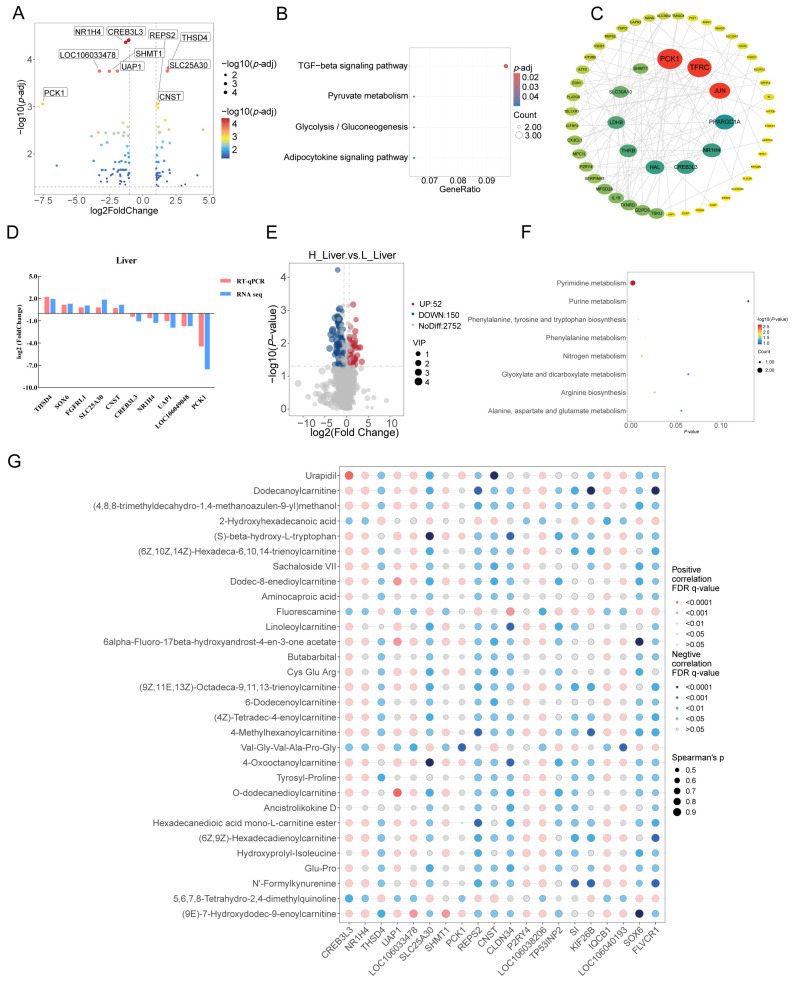
Transcriptomic and metabolomic analysis of the liver of Zhedong White geese of the HW and LW groups. (**A**) Volcano plot of DEGs in the liver of the HW and LW geese. (**B**) KEGG analysis of DEGs from the liver. (**C**) The protein–protein interaction network from hepatic DEGs between the two groups. (**D**) Validation of DEGs from liver by quantitative real-time PCR. (**E**) Volcano plot of DAMs in the liver of the HW and LW geese. (**F**) Functional analysis of DAMs in the liver. (**G**) The conjoint analysis of DEGs and DAMs in the liver.

**Figure 3 animals-15-01341-f003:**
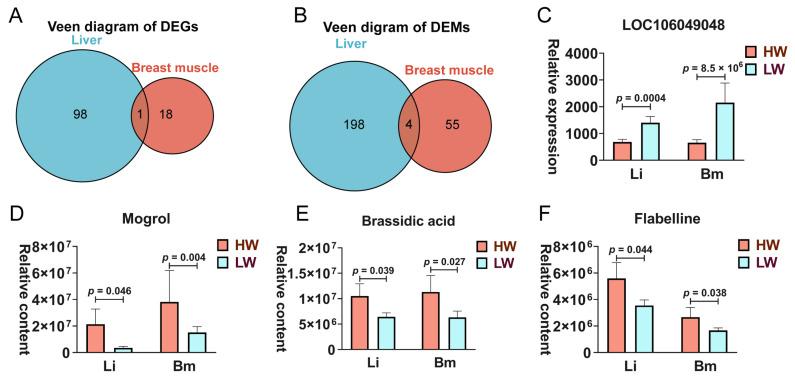
The conjoint analysis of DEGs and DAMs in the breast muscle and liver. (**A**) The Venn diagram of DEGs from the breast muscle and liver. (**B**) The Venn diagram of DAMs from the breast muscle and liver. (**C**) The level of LOC106049048 in the breast muscle and liver. The content of mogrol (**D**), brassidic acid (**E**), and flabelline (**F**). Li: liver; Bm: breast muscle.

**Table 1 animals-15-01341-t001:** The nutritional level of the geese diets.

Nutrition Composition	Geese Diets	
	0–28 Days	29–70 Days
Ingredients, %		
Corn grain	58.96	66.03
Soybean meal	26.67	15.81
Bran	2.30	2.10
Alfalfa meal	7.10	11.05
Soybean oil	1.85	2.00
Limestone	0.65	0.50
CaHPO_4_	1.06	1.10
L-lysine HCl	0.11	0.21
DL-methionine	0.10	0.00
NaCl	0.20	0.20
Premix ^1^	1.00	1.00
Total	100.00	100.00
Nutritional level, %		
Metabolic energy, Mcal/kg	2.90	3.00
Crude protein	19.00	15.00
Calcium	0.65	0.60
Digestibility phosphorus	0.30	0.30
Lysine	1.00	0.85
Methionine + cysteine	0.60	0.50

^1^ Provided per kilogram of diet: Cu (CuSO_4_·5H_2_O), 8 mg; Fe (FeSO_4_·7H_2_O), 80 mg; Zn (ZnSO_4_·7H_2_O), 90 mg; Mn (MnSO_4_·H_2_O), 70 mg; Se (NaSeO_3_), 0.3 mg; I (KI), 0.4 mg; Vitamin A, 9000 IU; Vitamin D3, 1600 IU; Vitamin E, 20 mg; Vitamin K3, 2 mg; Vitamin B11, 1.5 mg; Vitamin B2, 4 mg; Vitamin B6, 2 mg; niacin, 15 mg; folic acid, 0.6 mg; D-pantothenic acid, 10 mg; Vitamin B12, 0.02 mg; biotin, 0.13 mg; choline, 1000 mg.

**Table 2 animals-15-01341-t002:** The primers information.

Gene	Accession No.	Primer Sequence (5′–3′)
*COL11A2*	XM_066986897.1	catccagctgcccaagaaga
		ctgcttgagggagttgaggg
*COL22A1*	XM_066990776.1	aggactgaggcacaaagagc
		cactgtatcgcaccacacct
*TF*	XM_013186329.3	ggaccccaaaaccaaatgcc
		catccagacacagcagctca
*SYNC*	XM_048049583.2	ggcgactacttccaggagtg
		gcactccttcgtcaccttga
*MBOAT4*	XM_013196938.3	gttgcaaagctcctctaccg
		tcaaggtagcacaggacagg
*ANGPTL4*	XM_048077583.2	cttcaggcagctacccttct
		atggtggtggacttcagagg
*THSD4*	XM_013177461.3	gctgaattgccgtgccatag
		ccagacacaaccttgcaagc
*SOX6*	XM_066997629.1	gctttccctgacatgcacaa
		aggtacgttttggtcgaggt
*FGFRL1*	XM_048048078.2	aggttccgaatccttcagca
		acctggctgttctttcctga
*SLC25A30*	XM_066988573.1	tggaatgatgcatgcactgg
		ctcccgaaagaatgccacac
*CNST*	XM_066995523.1	aaaagagacagctggggagc
		tcgtcatcatcatcgggctg
*CREB3L3*	XM_048077817.2	ccagaaccaagagctgcaga
		ggacctggagaaaactcgca
*NR1H4*	XM_048061114.2	ccatgttcctccgttcagct
		agcgcgtattcttcctgtgt
*UAP1*	XM_067001427.1	atcgggttctgcttggagaa
		cggtggtgaagaagtggttg
*LOC106049048*	XM_048054800.2	aggcttgccggtcatagttc
		cgggttccagttttgcagtg
*PCK1*	XM_013190722.3	gcagccatgagatctgaagc
		ttttctccatagccaggcca
*GAPDH*	XM_067004670.1	gagggtagtgaaggctgctg
		accatcaagtccaccacacg

**Table 3 animals-15-01341-t003:** Growth performance and carcass yield of Zhedong White geese.

Item	LW	HW
BW at D1 (g)	132.25 ± 9.29	127.25 ± 1.71
BW at D70 (g)	3579.25 ± 269.53 ^b^	4328.00 ± 47.83 ^a^
Average daily gain (g)	49.24 ± 3.98 ^b^	60.01 ± 0.68 ^a^
Percentage of half-eviscerated weight (%)	75.18 ± 8.02	69.96 ± 2.35
Percentage of eviscerated weight (%)	88.63 ± 8.51	84.15 ± 2.26
Head yield (%)	4.39 ± 0.41	4.17 ± 0.16
Brain yield (‰)	2.45 ± 0.30	2.07 ± 0.14
Breast muscle yield (%)	7.48 ± 1.52	6.99 ± 0.95
Heart yield (‰)	6.90 ± 0.83	6.48 ± 0.21
Liver yield (%)	2.37 ± 0.66	3.15 ± 0.54
Gallbladder yield (‰)	1.0 ± 0.43	0.98 ± 0.34
Gizzard yield (‰)	5.25 ± 0.43	5.39 ± 0.41
Glandular stomach yield (‰)	5.04 ± 0.71	6.25 ± 2.40
Pancreas yield (%)	3.36 ± 0.14	4.01 ± 0.63

Data indicate the means of four individuals per group. ^a,b^: Values within the same row with different superscripts differ significantly (*p*-value < 0.05).

## Data Availability

The sequencing data used in this study are available at the China National Center for Bioinformation (https://www.cncb.ac.cn/ (accessed on 8 October 2024)) and under Genome Sequence Archive (GSA) CRA019513 (RNA-seq data). The metabolomic sequencing data used in this study are available from https://doi.org/10.6084/m9.figshare.28822766.v1.

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
