# Peer review of "Integrative Transcriptomic and Metabolomic Analysis of Muscle and Liver Reveals Key Molecular Pathways Influencing Growth Traits in Zhedong White Geese"

_animals, 2025, doi:10.3390/ani15091341_

Round 1
Reviewer 1 Report
Comments and Suggestions for Authors
Overall, this manuscript primarily presents a descriptive account of the experiments, without providing in-depth analysis of the underlying biological mechanisms responsible for the differences between high- and low-body-weight geese. Within the population of 500 individuals, it is an evident observation that the top 10% heaviest geese (“fat geese”) have a higher lipid content and significantly different breast muscle and liver proportions compared to the bottom 10% lightest geese (“lean geese”). However, the authors did not explore the underlying causes of these differences, such as potential associations with feed intake, disease status, or other physiological factors—issues that should be clearly addressed in the main text.
Fortunately, the study identified genes involved in lipid and carbohydrate metabolism. It is therefore recommended that the authors investigate these pathways more thoroughly to elucidate the molecular basis of the observed phenotypic differences. In addition, the manuscript would benefit from the inclusion of visual and quantitative phenotypic comparisons between fat and lean geese, such as body condition images and analyses of muscle and liver characteristics (e.g., muscle fiber content and density, hepatic lipid droplet accumulation). Finally, the authors should clearly justify the rationale for performing transcriptomic and metabolomic analyses on both liver and breast muscle tissues. A deeper comparative and integrative analysis of these two tissue types is needed, rather than a superficial listing of differentially expressed genes.
- The author needs to revise the overall formatting of the manuscript, such as standardizing the paragraph format.
- Line 42, provide additional details for correlation analysis in the “2. Materials and Methods” section.
- In "2.1 Samples", were the 500 Zhedong White geese from the same batch? Please include the relevant information.
- In "2.1 Samples", provide further details regarding the body weight measurement process. Was the weight taken after a fasting period?
- Regarding the selection criteria for the geese used for slaughter, were they chosen from the top 10% heaviest and the bottom 10% lightest individuals? Additionally, how can it be confirmed that the lightest 10% were of normal growth, not diseased or previously diseased? This information should be included. What was the selection basis? Are the geese selected for slaughter full-sibling or half-sibling related?
- In Table 3, why is the standard deviation for BW at D1 and BW at D70 so small for individuals in the HW group? Please specify the number of individuals in each group.
- In Table 3, terms such as "Average daily gain", "Percentage of half-eviscerated weight", "Percentage of eviscerated weight", "Head yield", "Brain yield", "Breast muscle yield", "Heart yield", "Liver yield", "Gallbladder yield", "Glandular stomach yield", and "Pancreas yield" should be defined in the “2. Materials and Methods” section.
- In "2. Materials and Methods", please include all relevant software versions and references.
- In line 118, the genomic version used in this study is the 2015 scaffold genome; why was the latest 2024 T2T goose genome not used?
- The author should supplement sections 3.2 and 3.3 with additional information, such as DEGs, DAMs, and their associated enriched pathways, to enhance the credibility of the article.
- In lines 325-326, “The metabolomic sequencing data used in this study are available from the corresponding author upon reasonable request.” The corresponding author should proactively provide a link or other information to allow for verification of the metabolomic sequencing, rather than including a statement without providing further details.
- Further details are needed regarding the muscle fiber composition and biochemical indicators of liver fat.
Author Response
Dear Reviewer,
We are grateful for the insightful suggestions you provided regarding our study. We have found your comments to be of great value and have taken them into consideration to enhance the quality of our paper. After careful deliberation, we have made the necessary revisions, which we believe will be satisfactory.
The revised sections of the paper are highlighted in yellow for your convenience. Below, we outline the principal corrections made to the paper along with our responses to your specific comments.
We appreciate your time and effort in reviewing our work and hope that the revisions address your concerns effectively. Thank you once again for your constructive feedback.
Best regards
Q1: The author needs to revise the overall formatting of the manuscript, such as standardizing the paragraph format.
A1: We have carefully examined the paragraph format and revised the entire manuscript accordingly.
Q2: Line 42, provide additional details for correlation analysis in the “2. Materials and Methods” section.
A2: We have rewritten the details of the correlation analysis in line 42.
Q3: In "2.1 Samples", were the 500 Zhedong White geese from the same batch? Please include the relevant information.
A3:The 500 Zhedong White geese used in this study were raised in the same batch. They were all hatched and raised under identical conditions at farm to ensure consistency in feeding regimen and environmental exposure. This standardized approach was designed to minimize confounding variables and allow us to more accurately assess the specific factors under investigation in our research. The corresponding content was written in lines 93-95.
Q4: In "2.1 Samples", provide further details regarding the body weight measurement process. Was the weight taken after a fasting period?
A4: Body weight was recorded after a 12-hour fasting period, and this detail has been incorporated into the latest manuscript.
Q5: Regarding the selection criteria for the geese used for slaughter, were they chosen from the top 10% heaviest and the bottom 10% lightest individuals? Additionally, how can it be confirmed that the lightest 10% were of normal growth, not diseased or previously diseased? This information should be included. What was the selection basis? Are the geese selected for slaughter full-sibling or half-sibling related?
A5: Firstly, during the growth process, we promptly eliminated any geese that showed signs of illness. At 70 days of age, after fasting and weighing, we selected geese from the top 10% and bottom 10% in weight for further research. Therefore, the lighter geese were not diseased. Secondly, the geese used in this study were randomly selected from the hatchery at a young age, and the breeder geese were randomly mated, so genetic relationships of the 500 geese were not specified.
Q6: In Table 3, why is the standard deviation for BW at D1 and BW at D70 so small for individuals in the HW group? Please specify the number of individuals in each group.
A6: We checked the original data and found that there was a handwriting error here. We have corrected it in the latest manuscript.
Q7: In Table 3, terms such as "Average daily gain", "Percentage of half-eviscerated weight", "Percentage of eviscerated weight", "Head yield", "Brain yield", "Breast muscle yield", "Heart yield", "Liver yield", "Gallbladder yield", "Glandular stomach yield", and "Pancreas yield" should be defined in the “2. Materials and Methods” section.
A7: We provided a detailed description of these methods in lines 97-108.
Q8: In "2. Materials and Methods", please include all relevant software versions and references.
A8: We have included the software versions and corresponding references.
Q9: In line 118, the genomic version used in this study is the 2015 scaffold genome; why was the latest 2024 T2T goose genome not used?
A9:The latest T2T reference genome of geese is for Taihu geese. As the geese used in our study were Zhedong White geese, we chose the 2015 scaffold genome version, which was specifically developed for Zhedong White geese.
Q10: The author should supplement sections 3.2 and 3.3 with additional information, such as DEGs, DAMs, and their associated enriched pathways, to enhance the credibility of the article.
A10:We have summarized the differentially expressed genes (DEGs) and differentially abundant metabolites (DAMs) in the breast muscle and liver, and uploaded them to the supplementary materials.
Q11: In lines 325-326, “The metabolomic sequencing data used in this study are available from the corresponding author upon reasonable request.” The corresponding author should proactively provide a link or other information to allow for verification of the metabolomic sequencing, rather than including a statement without providing further details.
A11:The original metabolomic sequencing data have been uploaded to the Figshare.
Q12: Further details are needed regarding the muscle fiber composition and biochemical indicators of liver fat.
A12: We appreciate your suggestion and recognize its potential to enhance our research. However, we did not collect samples for this specific analysis during our study. We plan to explore this in future research.

Reviewer 2 Report
Comments and Suggestions for Authors
Authors conducted a study on Zhedong White geese to reveal the molecular differences between large and low liveweight individuals.
Comments on the manuscript:
- there is no data on the health status of low-weight geese, consequently it can not be stated that those were helathy.
- Please provide information about housing and environmental conditions of raising experimental goose
- What was the procedure of slaughter?
- Were there any transport before slaughter or was it on site?
- "RNA purity and quality were measured": could you provide the result?
- "The total 20 μL reaction mixture consisted of 2 μL cDNA, 0.5 μL of each primer": What was the quantity for cDNA and primer as well?
- Please prove with data that GAPDH was a stable reference gene for the study.
- Please write in figure legend the name of abbreviations (Li, Bm)
Author Response
Dear Reviewer,
We sincerely appreciate the thoughtful suggestions you have provided regarding our study. We recognize the value and utility of your feedback in enhancing the quality of our manuscript. We have meticulously reviewed your comments and have made the necessary revisions, which we trust will be satisfactory. The revised sections of the paper have been highlighted in yellow for your convenience. Below, we outline the principal corrections made to the paper along with our detailed responses to your specific comments.
We are grateful for the opportunity to address your concerns and believe that these revisions have significantly strengthened our work. Thank you once again for your constructive feedback.
Best regards
Q1: There is no data on the health status of low-weight geese, consequently it can not be stated that those were helathy.
A1: During the growth process, we promptly eliminated any geese that showed signs of illness, although there is no data of health status of low-weight geese.
Q2: Please provide information about housing and environmental conditions of raising experimental goose.
A2: We have added detailed information on the housing and environmental conditions of the experimental geese in the "Samples" section.
Q3:What was the procedure of slaughter?
A3:We have detailed the slaughter procedure in the latest manuscript.
Q4:Were there any transport before slaughter or was it on site?
A4:All geese were slaughtered on site, eliminating the need for any transportation, this ensured minimal stress of the geese.
Q5:"RNA purity and quality were measured": could you provide the result?
A5:The RNA quality results were showed in the below table and figure, while these data were not displayed in manuscript.
Sample ID |
RIN value |
LW_Bm_1 |
8.5 |
LW_Bm_2 |
8.7 |
LW_Bm_3 |
8.4 |
LW_Bm_4 |
8.1 |
HW_Bm_1 |
8.5 |
HW_Bm_2 |
8.6 |
HW_Bm_3 |
8.1 |
HW_Bm_4 |
8.1 |
LW_Li_1 |
8.4 |
LW_Li_2 |
8.2 |
LW_Li_3 |
8.3 |
LW_Li_4 |
8.6 |
HW_Li_1 |
8.7 |
HW_Li_2 |
8.1 |
HW_Li_3 |
8.6 |
HW_Li_4 |
8.2 |
Q6: "The total 20 μL reaction mixture consisted of 2 μL cDNA, 0.5 μL of each primer": What was the quantity for cDNA and primer as well?
A6: We have revised the description of the qRT - PCR reaction mixture.
Q7:Please prove with data that GAPDH was a stable reference gene for the study.
A7:GAPDH is a common housekeeping gene and has also been widely used in various qRT-PCR experiments of geese (Tong, 2025, Poultry Science; Hu, 2022, Poultry Science), so we adopted this gene as the household management gene.
Q8:Please write in figure legend the name of abbreviations (Li, Bm)
A8:The figure legend has been updated to include the full names corresponding to the abbreviations (Li, Bm).

Round 2
Reviewer 2 Report
Comments and Suggestions for Authors
Authors improved their manuscript but some of my previous questions are still partially answered:
Q2: Please provide information about housing and environmental conditions of raising experimental goose. - The information is still missing eg. temparature, light - darkness, closed barn?......
Q6: "The total 20 μL reaction mixture consisted of 2 μL cDNA, 0.5 μL of each primer": What was the quantity for cDNA and primer as well? - the volume of the solution does not inform the reader about the weight of molecule. Eg. 2 μl cDNA solution contained 1 mg cDNA or 200 μg?
Q7:Please prove with data that GAPDH was a stable reference gene for the study. - If others used the same gene successfully in an other study it does not mean that it was stable for this goose growth study. Its expression level should be unaffected by experimental factors.
Author Response
Dear Reviewer,
Thank you for your suggestions to enhance our paper. We apologize for the unresolved issues in our previous submission and have now responded to them in detail. We hope our revised answers can clarify your concerns. The latest modifications in the manuscript are highlighted in yellow.
Best regards
Below are our detailed responses.
Q2: Please provide information about housing and environmental conditions of raising experimental goose. - The information is still missing eg. temparature, light - darkness, closed barn?......
A2: we provided detail raising information in the latest manuscript.
Q6: "The total 20 μL reaction mixture consisted of 2 μL cDNA, 0.5 μL of each primer": What was the quantity for cDNA and primer as well? - the volume of the solution does not inform the reader about the weight of molecule. Eg. 2 μl cDNA solution contained 1 mg cDNA or 200 μg?
A6: In our study, 1 μg of total RNA from each sample was reverse transcribed into cDNA. Subsequently, 2 μL of the resulting cDNA was utilized for the following qRT-PCR procedure. Moreover, the concentration of the primers has been appropriately adjusted in the latest version of the manuscript.
Q7:Please prove with data that GAPDH was a stable reference gene for the study. - If others used the same gene successfully in an other study it does not mean that it was stable for this goose growth study. Its expression level should be unaffected by experimental factors.
Q7: In the table below, we present partial results of the qRT-PCR, with GAPDH results highlighted in bold. These results demonstrate that GAPDH expression was stable between different groups.
Sample Name |
Target Name |
CÑ‚ value |
L-5_LI |
GAPDH-LI |
15.93 |
L-5_LI |
GAPDH-LI |
15.90 |
H-7_LI |
GAPDH-LI |
15.80 |
H-7_LI |
GAPDH-LI |
15.68 |
L-5_LI |
THSD4 |
25.28 |
L-5_LI |
THSD4 |
25.33 |
H-7_LI |
THSD4 |
22.92 |
H-7_LI |
THSD4 |
22.94 |
L-2_BM |
GAPDH-BM |
10.94 |
L-2_BM |
GAPDH-BM |
10.96 |
H-7_BM |
GAPDH-BM |
11.12 |
H-7_BM |
GAPDH-BM |
11.09 |
L-2_BM |
COL22 |
28.93 |
L-2_BM |
COL22 |
28.88 |
H-7_BM |
COL22 |
25.31 |
H-7_BM |
COL22 |
25.17 |
H: high body weight geese; L: low body weight geese.
Li : Liver; BM: Breast muscle.
